# Efficient Generation of Multiple Seamless Point Mutations Conferring Triazole Resistance in *Aspergillus fumigatus*

**DOI:** 10.3390/jof9060644

**Published:** 2023-06-02

**Authors:** Mariana Handelman, Nir Osherov

**Affiliations:** Department of Clinical Microbiology and Immunology, Sackler School of Medicine, Tel-Aviv University, Ramat-Aviv, Tel-Aviv 69978, Israel; mariana.handelman1@gmail.com

**Keywords:** *Aspergillus fumigatus*, CRISPR-Cas9, ARMS-PCR, pTel, genetic manipulation

## Abstract

*Aspergillus fumigatus* is a common human fungal pathogen that can cause a range of diseases. Triazoles are used to treat *A. fumigatus* infections, but resistance is increasing due to mutations in genes such as *cyp51A*, *hmg1* and overexpression of efflux pumps. Verifying the importance of these mutations is time-consuming, and although the use of CRISPR-Cas9 methods has shortened this process, it still relies on the construction of repair templates containing a selectable marker. Here, employing in vitro-assembled CRISPR-Cas9 along with a recyclable selectable marker, we devised a quick and easy way to effectively and seamlessly introduce mutations conferring triazole resistance in *A. fumigatus*. We used it to introduce, alone and in combination, triazole resistance-conferring mutations in *cyp51A*, *cyp51B* and *hmg1*. With the potential to seamlessly introduce genes imparting resistance to additional existing and novel antifungals, toxic metals, and environmental stressors, this technique can considerably improve the ability to introduce dominant mutations in *A. fumigatus*.

## 1. Introduction

*Aspergillus fumigatus* is one of the leading human fungal pathogens and the most common mold pathogen in humans. In humans, it can cause a wide range of diseases, from allergies to invasive infections with high mortality rates in immunocompromised patients [1]. To date, three major classes of antifungals are used to treat *A. fumigatus* infection: the triazoles, the polyenes, and the echinocandins [2]. Triazoles bind and inhibit Cyp51, a cytochrome P450 lanosterol 14-α-demethylase, catalyzing an essential step in ergosterol biosynthesis. Resistance to triazoles is a rising concern worldwide [3]. In *A. fumigatus*, the primary triazole resistance mechanisms are a result of mutations in *cyp51A,* encoding cytochrome P450 lanosterol 14-α-demethylase, as well as *hmg1,* encoding HMG-CoA reductase (one of the first steps in the ergosterol biosynthesis pathway), and overexpression of efflux pumps [1,2].

Verifying the importance of resistance-conferring mutations in *A. fumigatus* relies on their reintroduction into a susceptible strain followed by drug susceptibility testing [4]. This process is time-consuming as it involves generating a single repair template constructed of the target gene with the mutation, attached to a resistance marker cassette, with homologous flanking regions of ~1500 bp from each side. The advent of CRISPR-Cas9 gene-editing techniques has sparked a revolution in our capacity to manipulate the genomes of filamentous fungi [5]. These methods primarily harness the prowess of the programmable DNA endonuclease Cas9, in conjunction with a single guide RNA (sgRNA), to selectively identify and cleave precise DNA target sites. Recently, Cas12a (also known as Cpf1), a member of the class 2 type V-A CRISPR system family, has been harnessed for genome editing. Cas12a possesses several unique and attractive features, including the fact that it does not need a trans-activating CRISPR RNA (crRNA), allowing two or more crRNAs to be encoded in a multiplex single transcript [6]. In filamentous fungi, various CRISPR-Cas9 gene-editing techniques have been employed. These methods encompass the integration of Cas9 into the fungal genome, transient extrachromosomal expression using an AMA1-based plasmid, and the recently developed Cas9 Ribonucleoprotein (RNP) method. Cas9 RNPs entail the assembly of purified Cas9 protein with the sgRNA in vitro, followed by direct transfection into fungal protoplasts. The Cas9 RNP approach has been successfully implemented in several filamentous fungi, including *A. fumigatus*, *Fusarium oxysporum*, *Maganaporthe oryzae*, *Penicillium chrysogenum*, and *Mucor circinelloides* [5,7]. This method presents several advantages over integrated or plasmid-based delivery systems. These advantages include immediate cleavage activity, independence from host machinery for complex expression, and rapid depletion of the complex from the host cell following transformation. One notable advantage of Cas9 RNP in *A. fumigatus*, compared to traditional homologous recombination, is the ability to insert a selectable marker into a desired locus using homologous flanking regions of approximately 50 bp on each side, as opposed to the previously required ~1500 bp [8]. However, there are currently two limitations associated with Cas9 RNP technology in filamentous fungi. Firstly, the use of a selectable marker, such as hygromycin B (HYG), is necessary to effectively isolate the transformants from the population of protoplasts seeded on the transformation plate. Secondly, the current Cas9 RNP technology is primarily limited to gene deletion/disruption and the generation of gene fusions. To address these limitations, we devised a strategy that combines Cas9 RNP technology with the pTel-hygR linear plasmid. pTel-hyg^R^ is based on a linear plasmid with human telomeric sequences at both ends [9]. The plasmid linearizes when introduced into the fungal cell, and is maintained as an episome. pTel plasmids are an effective tool in the transformation of filamentous fungi (*Podospora anserina*, *Botrytis cinerea* and *Magnaporthe oryzae*), including in combination with CRISPR-Cas9 methods [9,10].

Here, we demonstrate a new use for the pTel-hyg^R^ plasmid as a valuable recyclable marker, in combination with CRISPR-Cas9 methodology, for seamless introduction and isolation of single and multiple resistance-conferring point-mutations in the pathogenic mold *A. fumigatus*.

## 2. Materials and Methods

**Media, Strains and Plasmids**. Strains were grown on YAG agar plates (0.5% yeast extract, 1% dextrose, 0.01M MgSO4, trace elements solution, vitamin mix, 1.5% agar) for 48 to 72 h at 37 °C. Asexual spores (conidia) were collected using 0.02% Tween solution. Broth microdilution experiments were performed in RPMI-MOPS (10% (*v/v*) RPMI x10, 3.45% (*w/v*) MOPS pH 7.0), droplet assays were performed on YAG agar plates (1.5% agar). Mycelium was prepared for transformation by static incubation in 15 mL SAB broth in Petri dishes for 16–20 h at 37 °C. *A. fumigatus* transformed protoplasts were plated on YPGS agar plates (2% yeast extract, 0.5% peptone, 2% D-glucose, 1M sucrose, 1.5% agar for plates or 0.7% for top, 250 μg/mL HYG, pH = 6). Voriconazole (VRC) (Sigma) was added to the agar plates, as needed. Transformation solutions include TRAFO 1 (0.6 M KCl, 50 mM CaCl_2_, 5 mM Tris-HCl pH 7.5, autoclaved), TRAFO 2 (40% PEG3350 in TRAFO 1, autoclaved), digestion solution (10 mL TRAFO1, 5% VinoTaste FCE (Novo Nordisk), filter-sterilized) and Cas9 Working buffer (20 mM HEPES, 150 mM KCl, pH 7.5, filter-sterilized). The *AkuB*^KU80^ non-homologous end joining mutant derived from patient isolate CEA10 [11] was used to generate the mutant strains described in this report (Appendix A). Strains were stocked in 25% glycerol solution and stored in −80 °C. pTel-hyg^R^ (Figure 1A) was a generous gift from Prof. Amir Sharon, Tel Aviv University.

**Construction of *A. fumigatus* repair templates**. The target genes were amplified, from strains in which these mutations were found. PCR was performed using Q5^®^ High-Fidelity 2X Master Mix (NEB), using the primers detailed in Appendix A. The entire PCR reaction was run on an agarose gel (1%), and the desired band was excised, cleaned with the Wizard^®^ SV Gel and PCR Clean-Up kit (Promega), and DNA concentration was assessed using Nanodrop (Thermo Fisher, Waltham, MA, USA).

**crRNA design, RNA duplex and Cas9.** Two crRNAs were designed for each gene (Appendix A), targeting the 5′ and 3′ areas of the gene, using EuPaGDT: a web tool tailored to design CRISPR guide RNAs for eukaryotic pathogens [12]. crRNAs, tracrRNA and Cas9 nuclease were ordered from Integrated DNA Technologies (IDT, Coralville, IA, USA).

**Protoplasting and transformation of *A. fumigatus***. The protoplasting and transformation protocol was adapted from the TRAFO based protocol [13] and CRISPR-Cas9 protocol [8]. In detail, 3 × 10^7^ *AkuB*^KU80^ conidia were inoculated into 30 mL of SAB broth, divided into two Petri plates and sealed with tape. Plates were incubated overnight (16–20 h) at 37 °C to form a mycelium. The next day, the mycelium was collected, drained using a sterile miracloth, and washed with fresh SAB broth. The mycelium was transferred into a 50 mL tube containing 10 mL of freshly made sterile digestion solution, and incubated horizontally at 30 °C, 70 rpm, for two hours to generate protoplasts. During this time, TRAFO 1 was chilled to 4 °C, top agar medium was autoclaved and kept in a warm bath (~48 °C) and TRAFO 2 was autoclaved and left to chill at room temperature. After two hours, protoplasts were filtered through sterile miracloth and the flow-through was transferred to ice. The flow-through was centrifuged at 3000 rpm for 5 min at room temperature, then discarded, and the pellet containing the protoplasts was resuspended in 10 mL cold TRAFO 1 by gentle pipetting. The tube was centrifuged again at 3000 rpm for 5 min at room temperature and the pellet resuspended in 1 mL of cold TRAFO 1 by gentle pipetting. Protoplasts were counted, adjusted to 5 × 10^6^/mL with TRAFO 1 (on ice) and used immediately for transformation. RNP complex (0.17 μL of 33 μM gRNA duplex 1, 0.17 μL of 33 μM gRNA duplex 2, 0.17 μL of 1 μg/μL Cas9 nuclease, 2.5 μL Cas9 Working buffer) was prepared and incubated for 5 min prior to use, then mixed with 105 μL of protoplasts, 0.5–1 μg repair template, 0.5–1 μg of pTel-hyg^R^, and 25 μL TRAFO 2, in a 15 or 50 mL tube. The mixture was incubated for 50 min on ice. Then, 250 μL of TRAFO 2 was added and mixed by gentle pipetting, followed by an additional incubation of 20 min at room temperature. 6 mL of top agar medium supplemented with 250 μg/mL HYG was added to the tube, gently pipetted, and poured onto 15 mL YPGS plates supplemented with 250 μg/mL HYG. Two mixtures without repair template or plasmid were prepared and plated onto YPGS plates supplemented or not supplemented with HYG, as negative and positive controls, respectively. One mixture with the plasmid, but without repair template, was prepared and plated onto a YPGS plate supplemented with HYG, as a positive control for pTel-hyg^R^ uptake. Plates were parafilm-sealed and incubated up-side down for three days at 37 °C.

**Selection and isolation of transformants**. After three days, 1.5 mL of 0.02% Tween was used to collect all conidia generated on the transformation plates, usually, conidia from 3–4 identical transformation plates were pooled. 10^3^–10^5^ conidia from pTel-hyg^R^ +repair template transformation plates were seeded on YAG agar plates with/without 0.25–2 μg/mL VRC, and incubated for three days at 37 °C. pTel-hyg^R^ transformed control conidia were seeded similarly. After three days, four VRC-resistant colonies from pTel-hyg^R^ +repair template and one VRC-susceptible colony from pTel-hyg^R^, were streaked on fresh YAG agar plates, and after two more days, conidia from a single colony were spread on a fresh YAG agar plate.

**Rapid genomic DNA extraction, ARMS-PCR and sequence verification**. From each isolate plate, conidia were collected using a loop dipped in sterile double-distilled water (DDW) and transferred into an Eppendorf tube containing 500 μL of DDW, until the liquid was murky-green. The tubes were frozen in liquid nitrogen for 15 min, then transferred immediately to 100 °C for 5 min. Tubes were vortexed vigorously for 15 s and conidial debris was pelleted by spin-down. The upper, clear, aqueous phase was taken for further use. PCR reactions were performed using Red Taq 5X Master Mix (Larova). Screening of the isolates for the presence of the desired mutations was done by Amplification Refractory Mutation System PCR (ARMS-PCR) [14]. ARMS-PCR primers, listed in Appendix A, were designed using the University of Southampton PRIMER1: primer design for tetra-primer ARMS-PCR website. The final mutated isolates from each gene transformation were verified for the presence of the desired mutation by Sanger sequencing.

**Recycling of the pTel-hyg^R^ plasmid**. After two transfers on YAG agar plates, the final isolates were tested on YAG supplemented with 350 μg/mL HYG, for verification of pTel-hyg^R^ plasmid loss.

**Azole susceptibility assays**. MIC (minimal inhibitory concentration) was determined by CLSI M38-A2 broth microdilution methodology. Briefly, voriconazole (VRC), itraconazole (ITC) or posaconazole (POS) (Sigma) were diluted in RPMI-MOPS and loaded into 96-well plates, conidia were diluted to 5 × 10^4^ conidia/mL and were loaded into the wells. Plates were incubated at 37 °C for 48 h, then the lowest concentration of triazole in which no fungal growth was seen (observed by inverted light microscope) was set as the MIC. Droplet susceptibility assay was performed by inoculation of 10^4^, 10^3^ 10^2^ or 10 conidia in 10 µL 0.002% Tween on the surface of YAG agar plates containing different concentrations of VRC and incubation for 48 h at 37 °C.

## 3. Results

**Efficient generation of seamless single point mutations Hmg1 F262del, Cyp51A G448S and Cyp51B G457S in *A. fumigatus* using the recyclable marker pTel-hyg^R^**. *AkuB*^KU80^ protoplasts were prepared and transformed as described above. The entire target gene (*hmg1*, *cyp51A*, *cyp51B*) was replaced by a mutated version (Hmg1 F262del, Cyp51A G448S and Cyp51B G457S) using CRISPR/Cas9 technology (see Figure 1, as representative for all three mutations).

First, we assessed the contribution of the recyclable marker pTel-hyg^R^ in generating a high percentage of true mutants (Figure 2). *AkuB*^KU80^ protoplasts underwent CRISPR-Cas9-based transformation on YPGS plates, with or without repair template (Hmg1 F262del), under VRC selection (Figure 2A). Results show 2–4 colonies in the absence (Figure 2A.1) or presence (Figure 2A.2) of the repair template Hmg1 F262del. Sequencing of *hmg1* in the four colonies in Figure 2A.2 showed no F262del mutation (not shown). Next, we included pTel-hyg^R^ in the transformation in the absence (Figure 2A.3) or presence (Figure 2A.4) of the repair template Hmg1 F262del. Following HYG selection, we saw numerous transformants on both plates. To identify transformants that had undergone Hmg1 F262del mutation, conidia were collected from the *AkuB*^KU80^ +pTel-hyg^R^ (Figure 2A.3) and *AkuB*^KU80^ +pTel-hyg^R^ +Hmg1 F262del (Figure 2A.4) transformation plates and plated on YAG (10^5^/plate) supplemented with 0.5 μg/mL VRC, then incubated for three days at 37 °C (Figure 2B). *AkuB*^KU80^ +pTel-hyg^R^ conidia were not resistant to VRC (Figure 2B.1). *AkuB*^KU80^ +pTel-hyg^R^ + Hmg1 F262del repair template yielded ~200 VRC-resistant colonies, or ~0.2% of plated conidia (Figure 2B.2). Four isolates from plate 2B.2 were purified by passing twice on YAG, which led to the loss of pTel-hyg^R^ and HYG resistance (Appendix A). Mutations were verified by Sanger sequencing (Figure 2C). All four isolates displayed deletion of TCT (TTC TTC TCG to TTC TCG), leading to deletion of phenylalanine in *hmg1*. These results demonstrate the importance of using plasmid pTel-hyg^R^ for generating targeted *hmg1* point mutations at high frequency.

We used the same approach to generate the Cyp51A G448S and Cyp51B G457S mutant strains (Figure 3). CRISPR-Cas9 generated transformants were first selected on HYG (Figure 3A). Conidia were then collected and plated on VRC (Figure 3B) as described above. *AkuB*^KU80^ + pTel-hyg^R^ conidia without repair template were not resistant to VRC (Figure 3B.1). The addition of repair template Cyp51A G448S (Figure 3B.2) or Cyp51B G457S (Figure 3B.3) yielded many VRC-resistant colonies. Four isolates from each of these plates were purified by passaging twice on YAG, which led to the loss of pTel-hyg^R^ and HYG resistance (Appendix A).

To enable cheap, rapid, and high-throughput identification of the specific mutation in each isolate, we introduced ARMS-PCR (Figure 4). Briefly, three primer sets were used to detect the presence of a specific mutation in the sequence (Appendix A, Appendix A): OF and OR control primer set, amplifying the area around the mutation (cont), OF and WT amplifying the WT sequence if present, (WT), and OR and Mut primer set, amplifying the mutated sequence, if present (Mut). All four Cyp51A G448S (Figure 4A) and Cyp51B G457S (Figure 4B) isolates generated the mutated band pattern (control and Mut PCR bands with or without the WT band), while the control WT generated control and WT PCR bands but not the Mut band. The four Cyp51A G448S (Figure 4C) and Cyp51B G457S ARMS-PCR-verified isolates were subsequently purified by passing twice on YAG, which led to the loss of pTel-hyg^R^ and HYG resistance (Appendix A). They were then sequenced to confirm the mutation (Figure 4D). All showed the GGT to AGT mutation (G448S) in *cyp51A* and GGC to AGC mutation (G457S) in *cyp51B*. Together, these results demonstrate that our protocol using a recyclable selectable marker and repair template generated by PCR amplification of the mutated gene can rapidly and efficiently generate single, seamless point mutations conferring triazole resistance in *A. fumigatus*.

***A. fumigatus* strains Hmg1 F262del, Cyp51A G448S and Cyp51B G457S are triazole resistant**. We determined the triazole MICs of the Hmg1 F262del, Cyp51A G448S, and Cyp51B G457S strains by CLSI M38-A2 broth microdilution methodology (Table 1). Importantly, all four verified isolates from each mutant strain grew normally on YAG agar and gave identical MICs, so only the results of a representative strain are shown in Table 1. Results show that the Hmg1 F262del and Cyp51A G448S mutations confer strong ≥8-fold resistance to VRC, ITC, and POS, in line with previous work [4,13]. The Cyp51B G457S mutation confers moderate 4-fold resistance to VRC, and no resistance to ITC and POS, as previously described [15]. Droplet susceptibility assay on YAG agar plates (Figure 5) largely corroborated the broth microdilution results (VRC MICs of the *AkuB*^KU80^ = 0.25 µg/mL; Hmg1 F262del = 1 µg/mL, Cyp51A G448S = 2 µg/mL, and Cyp51B G457S = 1 µg/mL).

**Efficient generation of multiple seamless point mutations to generate strains Cyp51B G457S/Hmg1 F262del, and Cyp51B G457S/Hmg1 F262del/Cyp51A G448S in *A. fumigatus*.** We reasoned that several point mutations may be introduced by repeatedly transforming a strain, starting with the weakest resistance mutation under low VRC selection and progressing to gradually stronger resistance mutations under higher VRC selection steps. To generate the Cyp51B G457S/Hmg1 F262del double mutant strain, the Cyp51B G457S mutant was transformed with the Hmg1 F262del repair template. Transformants were first selected on HYG. Conidia were then collected and plated on VRC (Figure 6A). Cyp51B G457S +pTel-hyg^R^ conidia without Hmg1 F262del repair template were not resistant to 0.75 μg/mL VRC (Figure 6A.1). Addition of repair template Hmg1 F262del yielded many VRC-resistant colonies (Figure 6A.2). *Hmg1* mutations were purified by passing twice on YAG, which led to the loss of pTel-hyg^R^ and HYG resistance (Appendix A). Then they were verified by Sanger sequencing in four isolates (Figure 6B). All displayed deletion of TCT (TTC TTC TCG to TTC TCG) leading to phenylalanine deletion in *hmg1*. To generate the Cyp51B G457S/Hmg1 F262del/Cyp51A G448S triple mutant strain, the double mutant Cyp51B G457S/Hmg1 F262del strain was transformed with the Cyp51A G448S repair template. Cyp51B G457S/Hmg1 F262del + pTel-hyg^R^ conidia without repair template were not highly resistant to 1.75 μg/mL VRC (Figure 6C.1). Addition of repair template Cyp51A G448S yielded several strongly VRC-resistant green colonies (Figure 6C.2). *Cyp51A* mutations were purified by passing twice on YAG, which led to the loss of pTel-hyg^R^ and HYG resistance (Appendix A). Then they were verified by Sanger sequencing in three isolates (Figure 6D). All showed the GGT to AGT mutation (G448S) in *cyp51A*.

***A. fumigatus* strains Cyp51B G457S/Hmg1 F262del, and Cyp51B G457S/Hmg1 F262del/Cyp51A G448S are highly pan-triazole resistant**. We determined the triazole MICs of the Cyp51B G457S/Hmg1 F262del, and Cyp51B G457S/Hmg1 F262del/Cyp51A G448S strains by CLSI M38-A2 broth microdilution methodology (Table 1) Results show that the combination of the Cyp51B G457S and Hmg1 F262del Hmg1 F262del mutations confers synergistic resistance to VRC but not POS. Combination of the Cyp51B G457S, Hmg1 F262del and Cyp51A G448S mutations confers synergistic resistance to POS and additive resistance to VRC. Droplet susceptibility assay on YAG agar plates (Figure 5) largely corroborated the broth microdilution results (VRC MICs of the Cyp51B G457S/Hmg1 F262del = 4 µg/mL, and of Cyp51B G457S/Hmg1 F262del/Cyp51A G448S > 8 µg/mL).

## 4. Discussion

In this report, we describe in detail a technique we developed to quickly and seamlessly introduce point mutations suspected or known to cause triazole resistance in *A. fumigatus*. Using a recyclable selection marker, the insertion of selectable markers into the genome, which can affect gene expression, is avoided. The recyclable selection marker can be used more than once, thereby introducing several point mutations into the same genome. Seamless mutations are introduced sequentially in order of increasing resistance, by increasing the concentration of drug used for selection at each step. A previous study describing CRISPR-Cas9 marker-free point mutation in *A. fumigatus* using single strand DNA repair template achieved successful point mutation in 8% of transformants [16]. Our experience described here, failed to generate successful point mutations using double stranded DNA repair template in the absence of the pTel-hyg^R^ plasmid for initial HYG selection. The addition of the pTel-hyg^R^ plasmid resulted in 100% efficient, seamless gene replacement. We propose that the pTel-hyg^R^ plasmid enriches the subpopulation of protoplasts that have also taken up the repair template, thereby increasing the proportion of mutated protoplasts. Further investigations are warranted to assess whether comparable high transformation rates can be achieved utilizing a linear hyg^R^ cassette or the AMA-1-hyg^R^ circular plasmid, instead of the pTel-hyg^R^ plasmid. One potential limitation of this study is that it exclusively focused on the *AkuB^KU8^*^0^ non-homologous end joining (NHEJ) null mutant, which was employed to enhance the integration of the repair template through homologous recombination. However, previous studies utilizing Cas9 ribonucleoprotein (RNP) technology in *A. fumigatus* strains with intact NHEJ have demonstrated gene deletion efficiencies of 75–90% when employing homology flanks of 50 bp [8]. Therefore, we anticipate comparable outcomes when utilizing the methodology described in this study.

While we report using our new methodology to seamlessly introduce point mutations causing triazole resistance, we believe it can be adapted to introduce any genes conferring a selectable gain of function. The main advantages of the method we describe are as follows: (i) seamless integration of the mutation is not accompanied by the integration of a selectable marker that can affect gene activity, transcript stability, etc.; (ii) design and construction of the repair template are simple as they involve merely amplifying the mutation-containing target gene, without having to generate a large multipart construct containing a selectable marker; (iii) it allows for the introduction of multiple resistance-conferring mutations in the same strain; (iv) it enables the identification of mutations that do not confer resistance, as these will fail to generate transformants growing under triazole selection; (v) in combination with ARMS-PCR methodology, we can screen many isolates quickly and easily, with high success rates of up to 100%, reducing sequencing cost and time. We have found this to be important in cases where mutations confer only weak triazole resistance.

In summary, by combining Cas9-gRNA in vitro assembly, double-stranded repair template and a recyclable marker, we have developed a simple technique for seamless replacement of resistance genes in *A. fumigatus*. Future studies will evaluate the potential of this system to seamlessly introduce genes conferring resistance to additional existing and novel antifungals, toxic metals, and environmental stressors.

## Figures and Tables

**Figure 1 jof-09-00644-f001:**
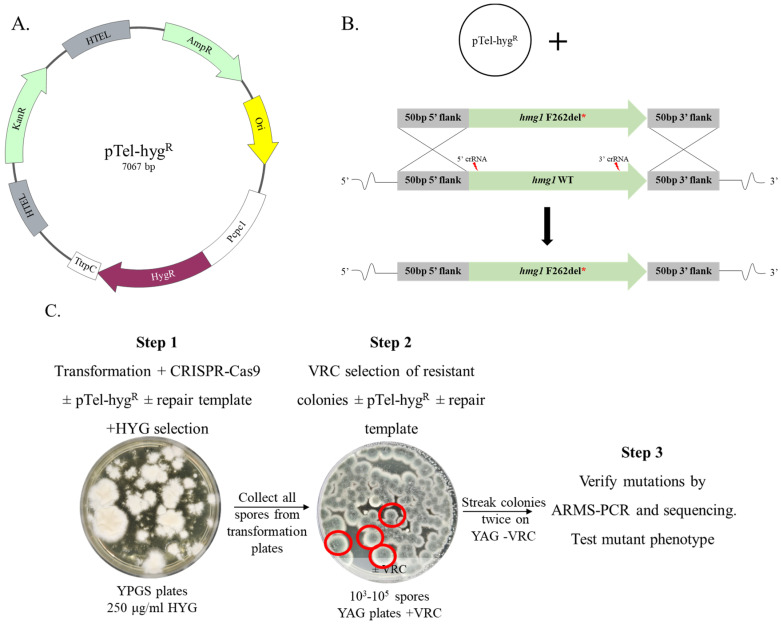
Outline of the seamless transformation approach developed for *A. fumigatus*. (**A**) Schematic of the pTel-hyg^R^ recyclable plasmid used for transformation (**B**) Gene replacement design. To introduce the *hmg1* point mutation, the PCR-amplified mutated *hmg1* gene was used as a repair template, transformation was performed with pTel-hyg^R^ and two guide crRNAs *hmg1* 5′ and *hmg1* 3’. Similar methodology was used to generate all gene replacements described. (**C**) Outline of the three experimental steps used to generate seamless point mutations. *A. fumigatus* protoplasts underwent CRISPR-Cas9-based transformation in the presence of the pTel-hyg^R^ recyclable plasmid and subsequent selection on HYG (step 1). Conidia were then further selected on VRC (step 2), streaked twice without HYG to remove pTel-hyg^R^, and (step 3) analyzed by ARMS-PCR and then by sequencing, to verify insertion of the mutation. Asterisk denotes mutated version of the gene. Red circles denote that these colonies were selected for ARMS-PCR and sequencing.

**Figure 2 jof-09-00644-f002:**
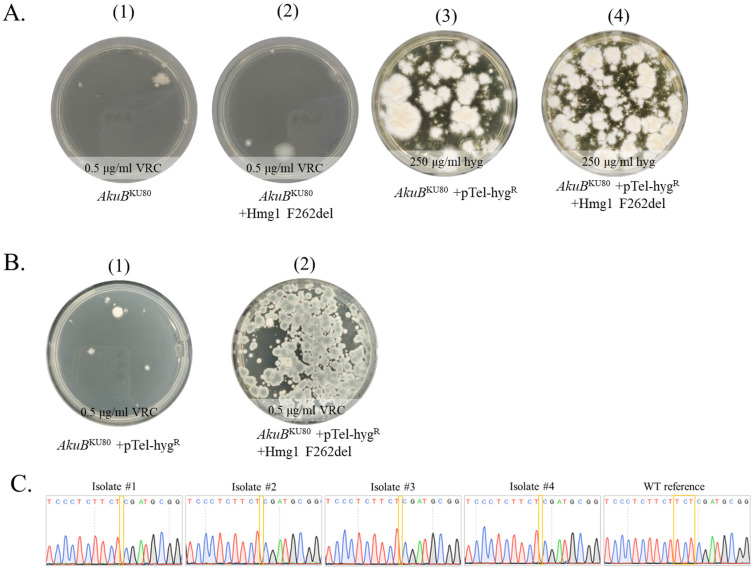
Generation of the Hmg1 F262del seamless mutation in *A. fumigatus*. (**A**) Primary selection of transformants on VRC (plates A1–A2) or HYG (plates A3–A4) with pTel-hyg^R^ (**B**) Secondary VRC selection of conidia collected from plates A3, A4 onto plates B1 and B2, respectively (**C**) Sequencing of *hmg1* in four colonies purified from plate B2, showing deletion of one of three TCT repeats leading to the F262del mutation, in comparison to the *AkuB*^KU80^ (WT) reference strain.

**Figure 3 jof-09-00644-f003:**
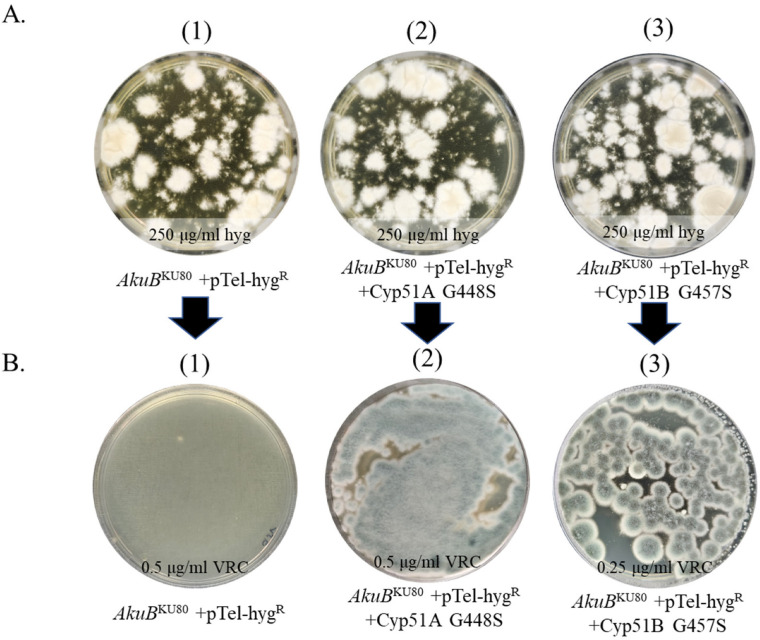
Generation of the Cyp51A G448S and Cyp51B G457S seamless mutations in *A. fumigatus*. (**A**) Primary selection of transformants with pTel-hyg^R^ on HYG (plates A1–A3) (**B**) Secondary VRC selection of conidia collected from plates A1–A3 onto plates B1–B3, respectively.

**Figure 4 jof-09-00644-f004:**
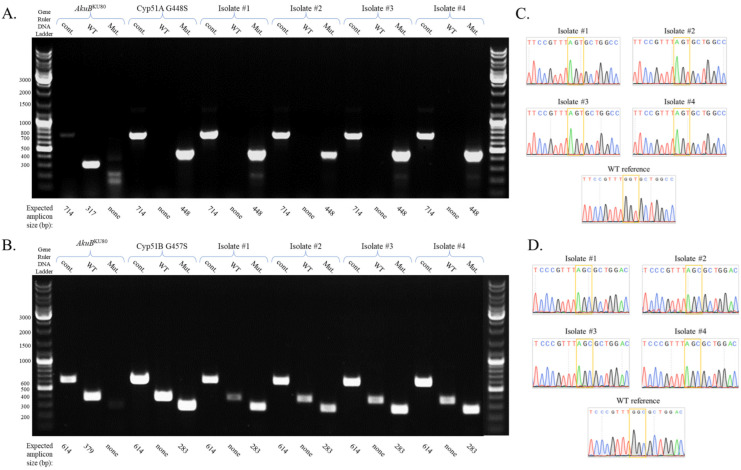
ARMS-PCR and Sanger sequencing verification of the Cyp51A G448S and Cyp51B G457S seamless mutations in *A. fumigatus*. ARMS-PCR analysis of four isolates transformed with (**A**) Cyp51A G448S WT or (**B**) Cyp51B G457S repair template. Strains were subsequently verified by Sanger sequencing (**C**) and (**D**) respectively, in comparison to the *AkuB*^KU80^ (WT) reference strain.

**Figure 5 jof-09-00644-f005:**
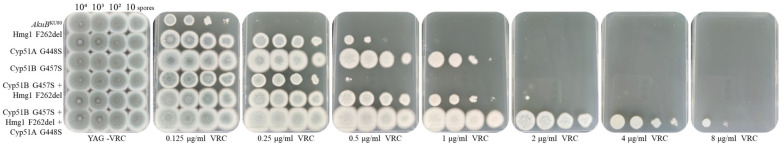
Droplet susceptibility assay of the strains generated in this study. Freshly harvested conidia were applied in serially diluted droplets onto YAG agar plates containing increasing concentrations of VRC. Images were taken after incubation for 48 h at 37 °C.

**Figure 6 jof-09-00644-f006:**
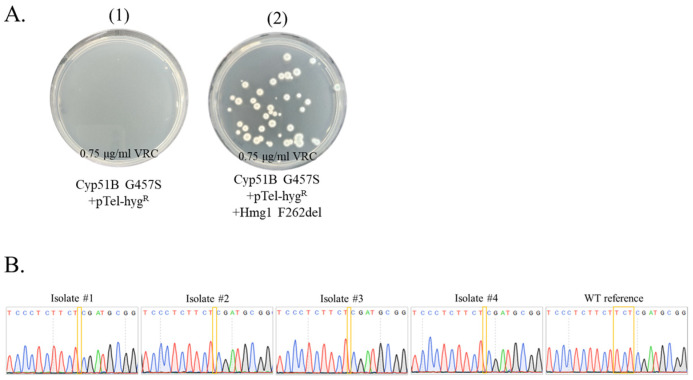
Generation and sequence verification of Cyp51B G457S/Hmg1 F262del and Cyp51B G457S/Hmg1 F262del/Cyp51A G448S strains. Secondary VRC selection of (**A**) Cyp51B G457S/Hmg1 F262del transformants and (**B**) sequence verification of selected isolates, and of (**C**) Cyp51B G457S/Hmg1 F262del/Cyp51A G448S transformants and (**D**) sequence verification of selected isolates.

**Table 1 jof-09-00644-t001:** MIC values for strains used in this study.

	MIC (μg/mL)		
Strain	VRC	ITC	POS
*AkuB* ^KU80^	0.5	0.5	0.03
Hmg1 F262del	4	>16	1
Cyp51A G448S	4	>16	0.5
Cyp51B G457S	2	0.5	0.03
Cyp51B G457S + Hmg1 F262del	16	>16	1
Cyp51B G457S + Hmg1 F262del + Cyp51A G448S	32	>16	4

## Data Availability

Not applicable.

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
