# Peer review of "Efficient Generation of Multiple Seamless Point Mutations Conferring Triazole Resistance in Aspergillus fumigatus"

_jof, 2023, doi:10.3390/jof9060644_

Round 1

Reviewer 1 Report

In the current manuscript, Handelman and Osherov described a rapid method to generate point mutations in the human pathogenic fungus Aspergillus fumigatus by using CRISPR RNP. The overall logic is clear, but this reviewer found the proper controls are missing in some figures. Additionally, the introduction and discussion are too brief and lack details. Please find my comments below.

Major concern:

One highlight of this work is the use of a recycle marker with telomeric repeats, as mentioned in the method on Line 136. However, this reviewer did not find/see any evidence to support that the vector was lost after streaking twice. It would be necessary to show data demonstrating the percentage of transformants that lose or retain resistance (in this case, hygromycin).

Minor concerns:

Line 28: The full name “Aspergillus fumigatus” should be used in its first appearance.

Line43: It is recommended to introduce more about CRISPR and RNP in the introduction, as these details are currently lacking. For example, there should be more information on how CRISPR facilitates gene manipulation in fungi. What is the benefit of using RNP instead of the traditional DNA-based vector? What is the current progress of using RNP in filamentous fungus? Is there any other way to make a site mutation (e.g., CRISPR-based editing or prime editing)?

Please see more information from the below references.

https://www.sciencedirect.com/science/article/pii/S2001037019300698

https://www.nature.com/articles/s41598-018-32702-w

https://www.nature.com/articles/s41467-022-34736-1

https://www.cell.com/trends/microbiology/fulltext/S0966-842X(23)00097-5

https://www.science.org/doi/full/10.1126/sciadv.abg2661

Fig1: The definition of the plus and minus signs in Fig1c is unclear.

Fig2: The content and presentation in fig2 as well as fig3, are confusing. It is unclear whether the CRISPR system was used in these assays. Additionally, It is awkward to see the shape of your phone in Fig2A.

Fig2C, Please include a sequencing result from the wild-type strains as a reference for comparison.

Fig4, this reviewer would like to suggest the authors include a diagram showing the location of each primer binding site and the expected results for different variants. For fig4c and d, please include a wild-type strain as the reference.

Fig5, please include a wild-type reference in Fig5b.

Line305, the statement "no other alterations in the target genome, such as the insertion of selectable markers, are generated" lacks evidence. If this is a highlight/focus of the manuscript, data from whole-genome sequencing is required to support it.

Figure 315-317, is that specific to pTel-hygR? How about any other fungal vector with hyg selection or even liner hyg fragment?

Minor editing of English language required

Author Response

In the current manuscript, Handelman and Osherov described a rapid method to generate point mutations in the human pathogenic fungus Aspergillus fumigatus by using CRISPR RNP. The overall logic is clear, but this reviewer found the proper controls are missing in some figures. Additionally, the introduction and discussion are too brief and lack details. Please find my comments below.

Major concern:

One highlight of this work is the use of a recycle marker with telomeric repeats, as mentioned in the method on Line 136. However, this reviewer did not find/see any evidence to support that the vector was lost after streaking twice. It would be necessary to show data demonstrating the percentage of transformants that lose or retain resistance (in this case, hygromycin).

REPLY:  All transformants described in the manuscript lost hygromycin resistance after 2 passages on YAG.  This data has been added in Figure S1 and is referred to in the text (L213, L235, L254, L297, L307)

Minor concerns:

Line 28: The full name “Aspergillus fumigatus” should be used in its first appearance.

REPLY: This has been corrected

Line 43: It is recommended to introduce more about CRISPR and RNP in the introduction, as these details are currently lacking. For example, there should be more information on how CRISPR facilitates gene manipulation in fungi. What is the benefit of using RNP instead of the traditional DNA-based vector? What is the current progress of using RNP in filamentous fungus? Is there any other way to make a site mutation (e.g., CRISPR-based editing or prime editing)?

REPLY: This has been addressed in an additional paragraph in the Introduction, Lines 43-69.

Fig1: The definition of the plus and minus signs in Fig1c is unclear.

REPLY: This has been corrected, the plus/minus signs have been removed.

Fig2: The content and presentation in fig2 as well as fig3, are confusing. It is unclear whether the CRISPR system was used in these assays. Additionally, It is awkward to see the shape of your phone in Fig2A.

REPLY: Yes, CRISPR-Cas9 was used in Fig. 2 and Fig. 3. This has been added (L177 and L206).  The phone reflections in 2A have been strongly reduced.

Fig2C, Please include a sequencing result from the wild-type strains as a reference for comparison.

REPLY: This has been added.

Fig4, this reviewer would like to suggest the authors include a diagram showing the location of each primer binding site and the expected results for different variants. For fig4c and d, please include a wild-type strain as the reference.

REPLY: A diagram showing the location of each primer binding site and the expected results for different variants has been added in the supplementary materials (Fig. S2) and has been mentioned in L246. Wild-type strain sequences have been added in the revised Fig. 4.

Fig6B, please include a wild-type reference.

REPLY: This has been added

Line305, the statement "no other alterations in the target genome, such as the insertion of selectable markers, are generated" lacks evidence. If this is a highlight/focus of the manuscript, data from whole-genome sequencing is required to support it.

REPLY: This has been corrected to “Using a recyclable selection marker, the insertion of selectable markers into the genome, which can affect gene expression, is avoided” L336

Figure 315-317, is that specific to pTel-hygR? How about any other fungal vector with hyg selection or even liner hyg fragment?

REPLY: We do not know. It would be interesting to try!

Reviewer 2 Report

In this study, the authors demonstrate a protocol to introduce point mutations in A. fumigatus exploiting CRISPR-Cas9 methodology and using pTel-hyg plasmid as a recyclable marker. The technique developed will be useful for the introduction of point mutations in the genes of interest, and the results are convincing. The minor comments are:

1.     In the method section, the authors mention TRAFO based protocol, indicating reference no. 10. In this reference, the protocol is unclear. It will be good if the authors brief about this protocol in this study.

2.     Mycelia were used for transformation; how they were counted to adjust them to 5 x 106 for transformation?   

Author Response

In this study, the authors demonstrate a protocol to introduce point mutations in A. fumigatus exploiting CRISPR-Cas9 methodology and using pTel-hyg plasmid as a recyclable marker. The technique developed will be useful for the introduction of point mutations in the genes of interest, and the results are convincing. The minor comments are:

  1. In the method section, the authors mention TRAFO based protocol, indicating reference no. 10. In this reference, the protocol is unclear. It will be good if the authors brief about this protocol in this study.

REPLY: The protocol has been explained in detail in lines 108-135

  1. Mycelia were used for transformation; how they were counted to adjust them to 5 x 106 for transformation?

REPLY: Mycelia were used to generate protoplasts. Protoplasts were then used for transformation. This is now made clearer on Line 113.

Round 2

Reviewer 1 Report

Most of my previous concerns have been addressed by the authors. While there are some minor contents that deserve further modification.

Line 49: Replace (REF) with the actual reference.

Line 51: Maintain consistency, use "Cas9" instead of "CAS9".

Line 52: Please use the full name "Ribonucleoprotein" (RNP) for its first appearance in the manuscript.

Line 54: Please consider introducing other CRISPR variants such as Cas12a for their applications in filamentous fungi, as they are a part of CRISPR editing system with increased interest.

Line71:  Please tune down your statement, similar based-editing approaches have been applied in filamentous fungi, as demonstrated in  https://pubs.acs.org/doi/full/10.1021/jacs.2c10211

Please discuss the potential effect of deficient NHEJ  due to the KU80 deletion mutant background strain used in the manuscript. Additionally, explore whether the approach can be applied to isolates with a functional NHEJ pathway.

Line 315: It is worth adding our below discussion in the MS.

"

"Figure 315-317, is that specific to pTel-hygR? How about any other fungal vector with hyg selection or even liner hyg fragment?

REPLY: We do not know. It would be interesting to try!"

"

Line 363, tune down your statement, it would be better to say "up to 100%" instead of "100%".

Author Response

Most of my previous concerns have been addressed by the authors. While there are some minor contents that deserve further modification.

Line 49: Replace (REF) with the actual reference.

REPLY: This REF is an error and is not needed. I removed it.

Line 51: Maintain consistency, use "Cas9" instead of "CAS9".

REPLY: Corrected and checked throughout manuscript

Line 52: Please use the full name "Ribonucleoprotein" (RNP) for its first appearance in the manuscript.

REPLY: Corrected

Line 54: Please consider introducing other CRISPR variants such as Cas12a for their applications in filamentous fungi, as they are a part of CRISPR editing system with increased interest.

REPLY: Thanks! I have added this (L47-51) plus reference 6.

Line71:  Please tune down your statement, similar based-editing approaches have been applied in filamentous fungi, as demonstrated in  https://pubs.acs.org/doi/full/10.1021/jacs.2c10211

REPLY: The statement has been toned down to “Secondly, the current Cas9 RNP technology is primarily limited to gene deletion/disruption and the generation of gene fusions.” L69

Please discuss the potential effect of deficient NHEJ due to the KU80 deletion mutant background strain used in the manuscript. Additionally, explore whether the approach can be applied to isolates with a functional NHEJ pathway.

REPLY: This has been added to the discussion, L353-360.

Line 315: It is worth adding our below discussion in the MS.

"Figure 315-317, is that specific to pTel-hygR? How about any other fungal vector with hyg selection or even liner hyg fragment?

REPLY: We do not know. It would be interesting to try!"

REPLY: This has been added in the discussion L351-3.

Line 363, tune down your statement, it would be better to say "up to 100%" instead of "100%".

REPLY: This has been corrected L372.